# SpecExec: Massively Parallel Speculative Decoding for Interactive LLM Inference on Consumer Devices

**Ruslan Svirschevski**[*†♡]    **Avner May**[*♣]    **Zhuoming Chen**[*‡]

**Beidi Chen**[‡◇]    **Zhihao Jia**[‡]    **Max Ryabinin**[♣]

† Yandex    ♡ HSE University    ♣ Together AI    ‡ Carnegie Mellon University    ◇ Meta AI

ruslansv@gmail.com, avner@together.ai, zhuominc@andrew.cmu.edu,
{zhihaoj2,beidic}@andrew.cmu.edu, mryab@together.ai

## Abstract

As large language models gain widespread adoption, running them efficiently becomes a crucial task. Recent works on LLM inference use speculative decoding to achieve extreme speedups. However, most of these works implicitly design their algorithms for high-end datacenter hardware. In this work, we ask the opposite question: *how fast can we run LLMs on consumer machines?* Consumer GPUs can no longer fit the largest available models and must offload them to RAM or SSD. With parameter offloading, hundreds or thousands of tokens can be processed in batches within the same time as just one token, making it a natural fit for speculative decoding. We propose SPECEXEC (Speculative Execution), a simple parallel decoding method that can generate up to 20 tokens per target model iteration for popular LLM families. SpecExec takes the most probable continuations from the draft model to build a "cache" tree for the target model, which then gets validated in a single pass. Using SpecExec, we demonstrate inference of 50B+ parameter LLMs on consumer GPUs with RAM offloading at 4–6 tokens per second with 4-bit quantization or 2–3 tokens per second with 16-bit weights. [1]

## 1 Introduction

Open-access large language models (LLMs), such as Llama [Touvron et al., 2023] and Mistral [Jiang et al., 2023], have become increasingly capable in the past years, and their adoption has grown dramatically. Although these models are openly available, users who are interested in running these models on consumer-grade GPUs (for example, due to privacy or cost reasons) face significant challenges. Many open-access LLMs are too large to fit on consumer GPUs, which necessitates offloading them onto CPU RAM to perform inference. Given the limited memory bandwidth between the CPU and the GPU, as well as the fact that all model parameters must be transferred to the GPU for the LLM to generate each new token, offloading is extremely slow and bandwidth-bound. For example, generating a single token using Llama 2-70B in 16 bit with offloading on an RTX 3090 GPU takes at least 4.5 seconds[2].

A recent line of work that aims to speed up LLM inference is speculative decoding [Leviathan et al., 2023, Chen et al., 2023a], which uses a small draft model to predict the next tokens and a larger target model to verify which of those tokens to accept in parallel. Although speculative decoding is a promising direction, the speedups that existing methods can attain in the offloading setting are relatively modest. While studying existing approaches [Leviathan et al., 2023, Miao et al., 2023, Sun et al., 2023], we discovered that these methods do not scale well with the draft model token budget. In particular, as shown in Figure 1 (left), the number of tokens accepted by the target model is

---

[*]Equal contribution.

[1]The code is available at `github.com/yandex-research/specexec`.

[2]Assuming PCIe-4.0 and at least 140GB of DDR5 RAM with an efficient offloading implementation.

38th Conference on Neural Information Processing Systems (NeurIPS 2024).

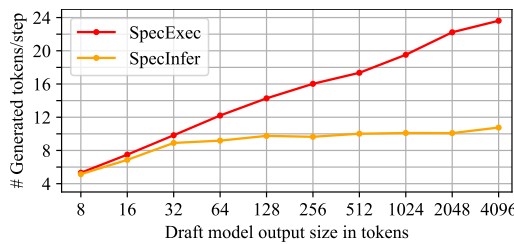 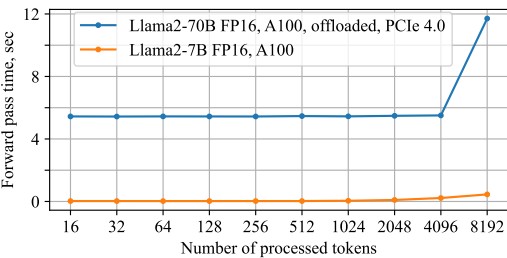

Figure 1: Acceptance counts vs draft size (left), forward pass GPU time vs input size (right). Llama 2-7B draft model, offloaded Llama 2-70B target model, MTBench dataset, t=0.6 and top-p=0.9.

empirically upper-bounded (approximately by 10 for this model and dataset combination) regardless of the number of speculated tokens. In turn, methods that scale better with more draft tokens Chen et al. [2024a] rely on static tree structures that may not be optimal for every setting, as they require tree structure optimization for every change in the text domain, generation parameters, and the hardware setup.

In this work, we aim to improve the effectiveness of speculative decoding for running large language models on consumer hardware with RAM offloading. We propose SpecExec, a speculative decoding method that addresses the performance, flexibility and scalability issues of prior methods. SpecExec[3] adopts a powerful draft model to deterministically[4] construct a large draft tree that covers the most likely continuations of the prefix with a parallel search algorithm. We then apply a simple verification algorithm that views this tree as a cache of potential continuations and validates it with the target model in a single pass.

Our main contributions can be summarized as follows:

1. We analyze the empirical behavior of speculative decoding algorithms with large language models and identify ways to improve their acceptance rate when scaling to thousands of draft tokens.

2. We propose SpecExec — a speculative decoding algorithm that improves the structure of generated draft trees for very large token budgets. We demonstrate that this technique can produce draft trees resulting in 10–20 accepted tokens with sufficiently large budgets.

3. Using our observations and SpecExec, we design a system that can run Llama 2-70B or comparable models interactively at 4–6 tokens/second using 4-bit quantization or 2–3 tokens/second with 16-bit weights on consumer GPUs with offloading, with 10–18x speedups compared to sequential inference on the same hardware.

## 2 Background

### 2.1 Speculative Decoding

In this study, we extend a family of algorithms for speculative decoding of autoregressive LLMs Stern et al. [2018], Leviathan et al. [2023], Chen et al. [2023a]. These algorithms generate tokens in two phases: *drafting* and *verification*.

During the drafting phase, the algorithm generates a candidate sequence of tokens by sampling from a small *draft model* $P(x_{t+1}|x_{0:t}, \theta_{\text{draft}})$. In turn, the verification stage leverages the *target model* $P(x_{t+1}|x_{0:t}, \theta_{\text{main}})$ to verify these draft sequences and accept all tokens that have passed the verification. The probability of accepting a token is chosen in a way that preserves the output distribution of sequential sampling from the original LLM Leviathan et al. [2023], Chen et al. [2023a], Sun et al. [2023]. A key advantage of speculative algorithms is that the main model can verify *all* draft tokens in parallel, which is more efficient than sequentially generating one token at a time.

---

[3]We chose this name because our method directly applies speculative execution to LLM inference. The draft model "guesses" which token prefixes the target model will need to continue, and then the target model computes distributions of continuations with a single forward pass on the speculated prefix tree.

[4]In contrast, speculative sampling requires stochastic generation of the draft tree using draft probabilities.

Subsequent works in speculative decoding extend this idea in several directions, including generating multiple draft sequences or draft trees, using multiple draft models, and finetuning the draft models to improve generation speed Miao et al. [2023], Liu et al. [2023], Xu et al. [2023]. Another line of follow-up studies explores alternative sources for the draft model: namely, self-speculative decoding uses a subset of main model layers to produce a draft Zhang et al. [2023], REST retrieves draft sequences from a search index He et al. [2023], and staged speculative decoding uses multiple levels of speculation Spector and Re [2023]. Leveraging these techniques, practitioners have built efficient implementations for fast LLM inference [Miao et al., 2023, Cai et al., 2023]. We refer the readers to survey papers for a more detailed coverage of speculative decoding methods Zhang et al. [2024a], Xia et al. [2024a].

In our analysis, we focus on speculative decoding algorithms that support sampling from the target model and guarantee identical sample probabilities vs standard generation. The rationale for our choice is that most popular LLM applications (such as chat assistants) require stochastic sampling to introduce variability into their responses. This focus rules out several algorithms that only support greedy inference Fu et al. [2023], Liu et al. [2023]. Still, most works on speculative decoding fit within that criterion.

## 2.2 Parameter Offloading

Another recent line of work explores running and training large models with limited accelerator memory by "offloading" their parameters to more abundant storage, such as RAM or even SSD [Pudipeddi et al., 2020, Ren et al., 2021, Alizadeh et al., 2023]. This technique works by loading model parameters on the GPU when they are needed for computation. Since most deep learning models use layers in a fixed order, offloading can pre-dispatch the next layer's parameters in the background.

This technique works particularly well when processing large batches of data, during training Pudipeddi et al. [2020], Ren et al. [2021] or large-batch non-interactive inference Aminabadi et al. [2022], Sheng et al. [2023], where each layer process multiple tokens each time it is loaded. In turn, when doing interactive inference, offloading works significantly slower than on-device inference. This is because interactive inference has to process one or few tokens at a time, and therefore spends most of the time waiting for the parameters to load.

## 2.3 Running LLMs on Consumer Devices

While our observations are not specific to any particular LLM, we focus on a practical case of running modern instruction-tuned models such as Llama-2-70B-chat Touvron et al. [2023] and Mixtral 8x7B Jiang et al. [2024]. To better estimate the target hardware setups, we study communities dedicated to running large models locally, such as LocalLlama [2023]. A popular[5] hardware configuration for running those models locally is a desktop or a cloud instance with a single consumer-grade GPU[6] with 12–24 GB VRAM, 4–8 CPU cores, 32–64 GB RAM, and a PCIe 3.0 or 4.0 x16 bus between CPU and GPU. Another popular setup is devices without a dedicated GPU, such as MacBooks with an ARM-based CPU, 16 GB RAM, and an SSD. While this survey may not be fully representative, it reveals popular setups that are not targeted by most speculative decoding research.

Running the largest language models in this setup requires either extreme compression or offloading. While it is possible to fit 70B+ models into consumer GPUs by compressing them to 1.5–2 bits per parameter Chee et al. [2023], Tseng et al. [2023], doing so causes significant accuracy losses that defeat the purpose of running large models Dettmers and Zettlemoyer [2022], Tseng et al. [2023]. Thus, practitioners with consumer-grade hardware may find it optimal to run 50B+ models with mild (e.g. 4-bit) quantization and offload parameters from GPU to RAM or SSD Alizadeh et al. [2023].

## 3  Preliminary analysis

Speculative decoding with offloading benefits from the fact that it is more efficient to process tokens in parallel than sequentially. In conventional inference, this is caused by the higher arithmetic intensity

---

[5]Based on popular hardware guides such as Dettmers [2023] as well as setups examples (see A, B, C, D)

[6]For example, RTX 4060 or 4090 desktops, T4 or A2 VMs.

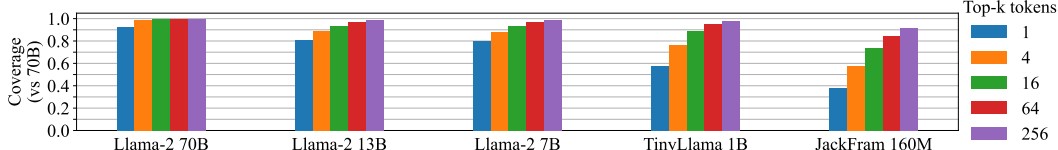

Figure 2: Llama-2 70B Chat model cumulative probability of most likely tokens compared to the draft model choice (all Llama draft models are chat versions), OASST1 dataset.

of GPU processing[7]. With offloading, there is a different bottleneck — loading model parameters from storage. Since offloading engines can dispatch model parameters in parallel with computation, the total processing time is the maximum of the time to load all parameters and the total computation time. In preliminary experiments (see Figure 1, right), we found that 70B models running on a consumer desktop can process thousands of tokens within nearly the same time as just a single token.

This leads us to a question: **how does speculative decoding perform when given hundreds to thousands of draft tokens?** As shown in concurrent work Chen et al. [2024a], speculative decoding algorithms with single or multiple sequences, like SpecInfer, are effectively upper-bounded in the number of accepted tokens as the speculation budget grows. This is confirmed by our observations (see Figure 1, left), where the number of accepted tokens saturates even for the more powerful Llama-2 7B.

In regular GPU inference, using 7B draft models would be impractical, as the drafting steps would take too long. However, in our setting, large draft models can be justified because each offloaded forward pass takes significantly more than a second (see Figure 1, right). This runs contrary to a popular intuition in speculative decoding that favors smaller draft models Miao et al. [2023], Liu et al. [2023].

We also observe that sampling from modern LLMs often results in a few high-probability tokens that add up nearly to a probability of 1 (see Figure 2 for an illustration). If we can find these tokens using the draft model, we can construct a draft that will be accepted with a similarly high probability. In preliminary experiments for 70B models, we found that running beam search with a capable draft model (e.g., Llama-2 7B) can recover many of these high-probability tokens. Unfortunately, this kind of deterministic search is incompatible with speculative decoding for stochastic sampling, which called for alternative validation method.

## 4 Method

### 4.1 Speculative Execution

As we observed in Section 3, high-probability continuations of large models are concentrated in a few tokens, and offloading benefits from running target model on hundreds or thousands of tokens. To use these observations, we formulate an alternative, simpler speculative decoding strategy. Unlike speculative decoding, SpecExec (short for "Speculative Execution") does not propose a new sampling procedure: it runs standard (sequential) sampling while trying to "guess" which probabilities will be needed during future steps and precomputing these continuations *in parallel*. This is similar to speculative execution [Lampson, 2006] in modern CPUs that predicts which operations should be computed ahead of time to better utilize the compute cycles.

More specifically, whenever SpecExec uses target model probabilities, it looks them up in a speculative "cache". If it encounters a token that is not in the cache, it queries the target model for that token and simultaneously computes probabilities for $B$ potential **future** tokens chosen with the draft model, where $B$ is the batch size. If the draft model can guess the likely next tokens accurately enough, the algorithm will be able to run multiple sampling iterations using these cached probabilities **without querying the target model** until it "exhausts the cache" and begins anew. A formal description of SpecExec is given in Algorithm 1.

To choose which future tokens should be precomputed, we run a search algorithm with the draft model to find $B$ most likely tokens according to their cumulative probability $\prod_t P(x_{t+1}|x_{0:t}, \theta_{\text{draft}})$.

---

[7]Parallel matrix multiplications do more useful computations per memory access for multiple tokens.

---

**Algorithm 1** SPECULATIVE EXECUTION

---

1: **Input:** prompt $x$, models $\theta_{\text{target}}, \theta_{\text{draft}}$, output length $L$, budget $K$, max depth $D$, batch size $B$
2: **Output:** a sequence of $L$ tokens generated by $\theta_{\text{target}}$
3: cache := PRECOMPUTE$(x, \theta_{\text{draft}}, \theta_{\text{target}}, K, D, B)$    ▷ target model probabilities for likely future tokens

4: **for** $t = 1, 2, \ldots, L$ **do**
5:    **if** $x \notin$ cache **then**
6:      cache := PRECOMPUTE$(x, \theta_{\text{draft}}, \theta_{\text{target}}, K, D, B)$
7:    $p_{\text{target}}$ := cache$[x]$      ▷ $p_{\text{target}}$ is equal to $P(\,\cdot\,|x_1, \ldots, x_t, \theta_{\text{target}})$
8:    $x_{\text{next}} \sim$ SAMPLE$(p_{\text{target}})$
9:    $x := x \oplus \{x_{\text{next}}\}$      ▷ append token
10: **return** $x$
11:
12: **function** PRECOMPUTE$(x, \theta_{\text{target}}, \theta_{\text{draft}}, K, D, B)$
13:    $\tau$ := CREATEDRAFTTREE$(x, \theta_{\text{draft}}, K, D, B)$    ▷ $\tau$ is a tree with $K$ tokens up to depth $D$
14:    next_probs := FORWARD$(\tau, \theta_{\text{target}})$    ▷ process $\tau$ tokens in parallel with offloading; note: next_probs is a matrix $\in \mathbb{R}^{K \times \text{vocab}}$
15:    cache := $\{\}$
16:    **for** $x_i \in \tau$ **do**
17:      $x_{\text{prefix}}$ := $\pi(x_i, \tau)$      ▷ prefix in tree $\tau$
18:      cache$[x_{\text{prefix}} \oplus \{x_i\}]$ = next_probs$[x_i]$    ▷ probabilities of possible next tokens
19:    **return** cache

---

The details of the search algorithm are given in Section 4.2; unlike drafting with regular speculative decoding, this procedure is deterministic and always selects tokens with the highest probability.

**Comparison to speculative decoding.** The core advantage of SpecExec over regular speculative decoding is that the algorithm does not need the draft tree to follow a known probability distribution. In other words, SpecExec produces correct samples with any draft tree, even if it is deterministic. We use this property to construct the best possible speculative tree in ways that would break the assumptions of standard speculative decoding. For instance, our tree construction procedure, outlined in Section 4.2, considers only the most likely draft tokens and aims to capture a larger portion of the total probability mass.

However, this advantage comes at the cost of lower acceptance rates for any individual token. Algorithm 1 accepts a token $x_t$ with probability $P(x_{t+1}|x_{0:t}, \theta_{\text{target}})$, because accepting a token with SpecExec is equivalent to sampling that token from the target model distribution. Meanwhile, the original speculative decoding (for example, Miao et al. [2023]) accepts tokens with a higher probability $P(x_{t+1}|x_{0:t}, \theta_{\text{target}})/P(x_{t+1}|x_{0:t}, \theta_{\text{draft}})$.

For a small number of draft tokens (for instance, just one token), SpecExec is less effective than traditional speculative decoding. However, as we increase the number of draft tokens, speculative execution generates better-structured trees, which in practice leads to accepting more tokens for the same draft size; we verify this in Section 5.2.

**Correctness.** Next, we need to verify that SpecExec is equivalent to sequential sampling from the target model. Notably, unlike Leviathan et al. [2023], SpecExec does not change the probabilistic sampling procedure. The difference between SpecExec and sequential decoding is that SpecExec precomputes some probabilities, thus improving the GPU utilization in the case of offloading.

From a formal perspective, we rely on the fact that a speculative generation algorithm is equivalent to sequential sampling if it is locally equivalent in every node Miao et al. [2023]; in other words, it samples from the same probabilities for every prefix in the draft tree. Since SpecExec explicitly samples from the same probabilities as the main model, this is true by construction.

The fact that SpecExec follows the same sampling procedure has another side effect. If we view SpecExec as a deterministic function that depends on a pseudo-random number generator as input, we can prove a stronger degree of equivalence. Namely, for every seed value of the random number generator, SpecExec produces exactly the same outputs as sequential sampling with the same seed. In

contrast, speculative decoding does not have this properly, as it only guarantees correct probabilities for the overall generation procedure.

## 4.2 Search for the Optimal Draft Tree

As we discussed above, our algorithm uses the draft model to build a tree $\tau$ of likely future tokens for speculative caching. In this section, we describe how to find these tokens efficiently. From an optimization perspective, we seek to construct a tree that will lead to the highest expected number of generated (accepted) tokens. This problem can be solved by viewing the tree construction as the search for the set of nodes (i.e., tokens) that have the highest cumulative probability with respect to the target model. As we show in Appendix A, this search can be reduced to the single-source shortest path (SSSP) search problem that can be solved efficiently using a modified version of the Dijkstra's algorithm Dijkstra [1959], described formally in Algorithm 2.

In summary, SpecExec follows a loop:

1. Run Algorithm 2 with the draft model to select $K$ best tokens,

2. Process them with the target model using offloading,

3. Follow Algorithm 1 to determine which tokens are accepted.

For a visual high-level representation of the SpecExec algorithm, see Appendix B.

While Algorithm 2 is a rather special case of SSSP over a combinatorially large tree (the tree of all token sequences up to length $K$), the general SSSP problem is well studied in the computer science community (see Appendix C). Therefore, practitioners will be able to leverage existing algorithms to implement Speculative Execution for a broad range of setups, including GPUs, mobile CPUs, or distributed systems.

---

**Algorithm 2** PARALLEL SSSP FOR DRAFTING

---

1: **Input:** prefix $x$, $\theta_{\text{draft}}$, budget $K$, depth $D$, batch $B$
2: **Output:** a tree of $K$ likely future tokens
3:
4: **function** CREATEDRAFTTREE($x$, $\theta_{\text{draft}}$, $K$, $D$, $B$)
5:   $\tau :=$ TREE($\{x\}$)       ▷ an empty tree with root at $x$
6:   $T := \infty$          ▷ stopping threshold
7:   $H :=$ PRIORITYQUEUE($\{x : 0\}$)   ▷ $x$ has priority 0; H is ordered by negative
                  cumulative log-probabilities
8:   **for** $d = 1, 2, \ldots, D$ **do**
9:    batch $:= \varnothing$
10:    **for** $b = 1, 2, \ldots, B$ **do**
11:     $H, x_b, nll_b :=$ EXTRACTMIN($H$)
12:     **if** $nll_b < T$ **then**
13:      $\tau :=$ ADDCHILD($\tau, x_b, nll_b$)
14:      batch $:=$ batch $\cup \{x_b\}$
15:    **if** batch $= \varnothing$ **then**
16:     **break**
17:    **if** SIZE($\tau$) $\geq K$ **then**
18:     $T := -$KTHCUMULATIVELOGPROB($\tau, K$) ▷ ignore tokens that fall outside the budget
19:    probs $:=$ FORWARD(batch, $\theta_{\text{draft}}$)    ▷ run $\theta_{\text{draft}}$ w/o offloading, attend to past tokens;
                    note: probs is a matrix $\in \mathbb{R}^{B \times \text{vocab}}$
20:    topk $:=$ SELECTBEST(batch, probs, $\tau, K$) ▷ select best tokens by cumulative probability
21:    **for** $(x_i, p_i) \in$ topk **do**
22:     $\log p_{\text{prefix}} :=$ CUMULATIVELOGPROB($x_i, \tau$) ▷ $\sum_{x_t \in \pi(x_i, \tau)} \log P(x_t | \pi(x_t, \tau, \theta_{\text{draft}}))$
23:     $nll := -\log p_{\text{prefix}} - \log p_i$
24:     $H :=$ INSERT($H, x_i, nll$)
25:    $H :=$ TRIM($H, K$)       ▷ remove all except K best
26:   **return** TRIM($\tau, K$)

---

### 4.3 Implementation Details

Finally, we leverage several important technical improvements that speed up inference in real-world conditions. When running the forward pass with an offloaded target model, we accelerate inference by loading the next layer parameters in parallel with computing the previous layer activations using a dedicated CUDA stream, which is known to speed up offloading in other use cases Pudipeddi et al. [2020], Ren et al. [2021], Aminabadi et al. [2022]. We also preload the first few layers of the target model on the GPU in the background while drafting for a further speedup. We describe additional implementation details in Appendix D.

In our experiments, we also consider quantizing target models using recent post-training quantization algorithms Frantar et al. [2022], Lin et al. [2023], Dettmers et al. [2023]. While quantization is generally popular among LLM practitioners, it is particularly useful for our use case, as quantized models take less time to load from RAM to GPU and have RAM offloading requirements attainable by consumer hardware.

## 5 Experiments

### 5.1 Probability Coverage

The core assumption behind Algorithm 1 is that a reasonably large draft can "cover" most of the high-probability sequences of the target model. This is only possible if the target model predictions have low entropy (i.e., there is a small number of tokens with high probability) and the draft model can guess these tokens most of the time.

To test these assumptions in isolation, we measure the total probability mass "covered" by $k$ most likely tokens, as well as the probability of top-$k$ tokens guessed by draft models of varying size. If a certain draft model achieves a coverage probability $p$ for $k$ tokens, this means that taking the $k$ most likely tokens predicted by the draft model and measuring their probabilities with the *main* model (Llama-2-Chat 70B) would result in an average cumulative probability equal to $p$. We evaluated multiple draft models of various size: JackFram-160M Miao et al. [2023], TinyLlama-1.1B-Chat v1.0 Zhang et al. [2024b], Llama-2-Chat 7B and 13B tou. We report these coverage probabilities on a sample of 512 OpenAssistant conversations Köpf et al. [2023]. For each conversation, we generate 64 additional tokens by sampling from the target 70B model probabilities. We sample these tokens using original probabilities (without temperature or top-p sampling) and use the same tokens for every draft model.

The resulting coverage is reported in Figure 2. This leads to several important observations. First, the target model (Llama-2 Chat 70B) tends to have sharp probability distributions, where the top 1–4 tokens cover 90–98% of the entire probability mass. This agrees with existing observations that language models (esp. the larger ones) are overconfident Miao et al. [2021], Chen et al. [2023b].

Next, we compare how effective the draft models are at predicting these high-probability tokens. While all models eventually get over 90% coverage rate, Llama-2 Chat 7B makes much more accurate predictions with the first 1–4 tokens. This is important for our use case because, while the full draft tree contains thousands of tokens, individual tokens within that tree have much fewer children. Curiously, the 13B draft model demonstrates roughly the same accuracy as 7B despite its larger size.

Though we evaluate coverage for "raw" probabilities from the 70B model, many practical inference scenarios use temperature or nucleus sampling Holtzman et al. [2020]. In fact, thedefault generation parameters for Llama-2 70B use both a temperature of 0.6 and top-0.9 nucleus sampling Face [2024]. Generating in this way makes the model even more confident, which further improves the efficiency of parallel decoding.

### 5.2 Draft Acceptance Rates

Next, we study how Speculative Execution compares to existing speculative decoding variants for different token budgets. Since all algorithms guarantee that the tokens are sampled from $P(x_{t+1}|x_{0:t}, \theta_{main})$, we compare them in their ability to generate longest sequences of accepted tokens given the same budget of draft tokens.

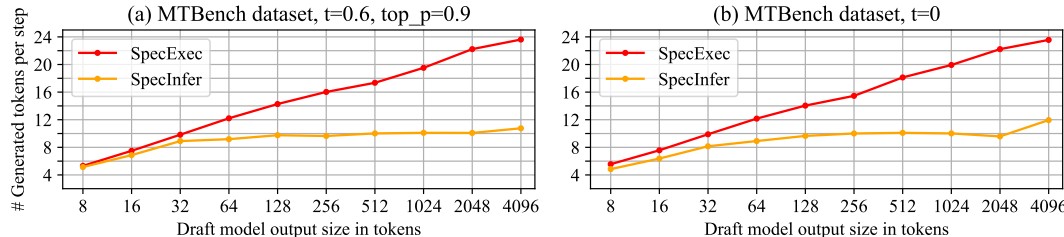

Figure 3: Generation rate depending on the draft budget size for Llama 2-7B Chat as the draft model and Llama 2-70B Chat as the target model, MTBench Zheng et al. [2023] dataset. Results are obtained with an A100 GPU.

Since we are interested in very large budgets, we choose baseline algorithms that better fit this task. The original speculative decoding algorithm Leviathan et al. [2023] generates a single sequence, which is truncated as soon as the algorithm rejects a single token. In other words, using a single long draft sequence results in most of the draft budget being wasted. Therefore, as our baseline, we choose the SpecInfer algorithm that shares the large token budget across multiple stems in a tree.

Similarly to the previous section, we use 70B versions of Llama 2 and Llama 2-Chat as target models. The draft model choice was driven both by the speed and the acceptance rate factors: we found that using draft models with 7B parameters results in significantly more accepted tokens, and a longer forward pass time is still affordable in the offloading setting. We report the effects of these draft models in more detail in Appendix E.

In each setup, we compared SpecExec and SpecInfer, using the 7B draft model, chosen based on our findings from Section 5.1. Figure 3 reports the average number of accepted tokens both for the default sampling configuration (temperature 0.6, top-p 0.9) and for greedy sampling for Llama 2-70B Chat model using the MTBench dataset. Similar tests were run for non-chat models on the C4 dataset; see Figure 4 for results.

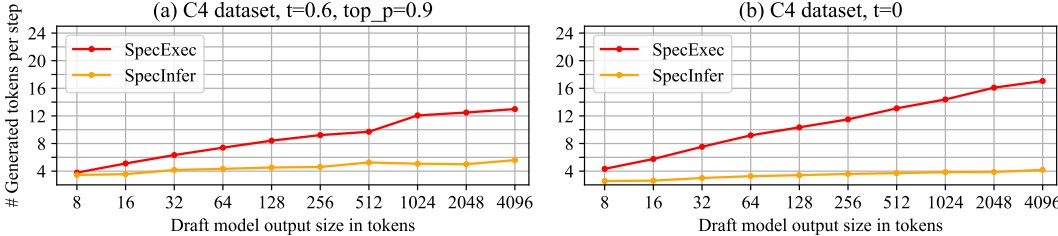

Figure 4: Generation rate depending on the draft budget size for Llama 2-7B Chat as the draft model and Llama 2-70B Chat as the target model, C4 dataset. Results are obtained with an A100 GPU.

For smaller draft budgets, SpecExec performs on par with SpecInfer, but eventually outscales it as we increase the number of draft tokens. We attribute this not to the general verification algorithm, but to the fact that SpecExec constructs its draft out of most likely tokens, while SpecInfer must sample draft tokens to guarantee correctness. We also observe that Speculative Execution achieves a higher margin of improvement on MTBench samples than on C4. It also accepts more tokens with a lower temperature. We attribute this to sharper token probability distributions, leading to higher coverage rates for the same number of draft tokens.

## 5.3 Inference Speed

Finally, we evaluate the practical inference speed of SpecExec by running it with offloading in different hardware configurations. We run these evaluations for Llama 2-70B tou models, both in regular and chat (instruction-tuned) versions. For prompts, we used subsamples of size 100 from OpenAssistant conversations Köpf et al. [2023], WikiText-2 Merity et al. [2016], MTBench Zheng et al. [2023], and C4 Raffel et al. [2020], measuring the speed of generating 32+ tokens per prompt. For Llama 2, we tested two setups: running the main model in 16 bits or quantizing it to 4 bits using GPTQ Frantar et al. [2022]. We also tested Mixtral 8x7B Jiang et al. [2024] (also quantized with GPTQ) and Llama 3 AI [2024] target models in fewer setups.

Table 1: Inference speed with RAM offloading, A100 GPU, Chat / Instruct models, using SpecExec (SX) and SpecInfer (SI) methods. Generation rate ("Gen. rate") denotes the average number of draft model tokens accepted for one target model iteration.

| Draft / Target models | Dataset | t | Method | Budget | Gen. rate | Speed, tok/s | Speedup |
|---|---|---|---|---|---|---|---|
| Llama 2-7B / 70B | OAsst | 0.6 | SX | 2048 | 20.60 | **3.12** | **18.7x** |
| | | 0.6 | SI | 1024 | 8.41 | 1.34 | 8.0x |
| | | 0 | SX | 1024 | 18.8 | **2.74** | **16.4x** |
| | | 0 | SI | 1024 | 7.86 | 1.18 | 7.1x |
| Llama 2-7B / 70B GPTQ | OAsst | 0.6 | SX | 128 | 12.10 | 6.02 | 8.9x |
| | | 0 | SX | 256 | 13.43 | 6.17 | 9.1x |
| Mistral-7B / Mixtral-8x7B | OAsst | 0.6 | SX | 256 | 12.38 | 3.58 | 3.5x |
| Llama 3-8B / 70B | | 0.6 | SX | 1024 | 18.88 | 2.62 | 15.6x |
| Llama 3-8B / 70B | MTBench | 0.6 | SX | 1024 | 18.16 | 2.79 | 16.6x |
| | | 0 | SX | 2048 | 21.58 | 2.94 | 17.5x |

We measure the inference speed with multiple GPU types: A100 (data-center GPU), RTX 4090 (current generation high-end consumer GPU), RTX 3090 (previous generation consumer GPU), and RTX 2080Ti (older consumer GPU). The first three GPUs are connected to the host via PCIe Gen 4 x16, while 3090 and 2080Ti were tested with PCIe Gen 3 x16. Note that for A100, we report the forward pass time with offloading, even though the GPU can fit a quantized model in its memory. We run all experiments with a batch size of 1 to match the setup of running LLMs on a local machine.

The average inference speed (measured in tokens per second) for A100 GPUs is reported in Tables 1 and 2. While the exact inference speed differs from setup to setup, Speculative Execution consistently speeds up generation with offloading by several times. These results compare favorably with recently published speculative decoding methods using fixed trees like Sequoia Chen et al. [2024b], which attains 2.2 tokens per second in the Llama 3-8B/70B setup, compared to 2.8 tokens per second in case of SpecExec.

In Table 3, we report the results of similar experiments for a range of real-world consumer GPUs. To reduce the memory requirements for the consumer setup, we replaced a 16-bit Llama-2 70B model with a 4-bit GPTQ compressed variant of Llama-2-70B as the target model. To lower the VRAM requirements for 2080 Ti, we used Sheared-Llama-1.3B Xia et al. [2024b] as a draft model, making the whole experiment consume just over 7 GB of VRAM. Note that while the fastest inference time is achieved on RTX 4090, slower consumer GPUs (for example, RTX 2080Ti) still generate tokens quickly enough for interactive use.

Table 2: Inference speed with RAM offloading. A100 GPU, base models, using SpecExec (SX) and SpecInfer (SI). Generation rate ("Gen. rate") denotes the average number of draft model tokens accepted for one target model iteration.

| Draft / Target models | Dataset | t | Method | Budget | Gen. rate | Speed, tok/s | Speedup |
|---|---|---|---|---|---|---|---|
| Llama 2-7B / 70B | C4 | 0.6 | SX | 2048 | 12.9 | **1.97** | **11.8x** |
| | | 0.6 | SI | 1024 | 6.48 | 1.03 | 6.2x |
| | | 0 | SX | 2048 | 16.1 | **2.38** | **14.3x** |
| | | 0 | SI | 1024 | 4.78 | 0.75 | 4.5x |
| Llama 2-7B / 70B | WikiText-2 | 0.6 | SX | 2048 | 9.57 | **1.54** | **9.2x** |
| | | 0.6 | SI | 1024 | 4.69 | 0.77 | 4.6x |
| | | 0 | SX | 2048 | 11.74 | **1.88** | **11.3x** |
| | | 0 | SI | 1024 | 3.71 | 0.62 | 3.6x |
| Llama 2-7B / 70B GPTQ | WikiText-2 | 0.6 | SX | 256 | 6.99 | 3.72 | 5.5x |
| | | 0 | SX | 256 | 8.81 | 4.54 | 6.7x |
| Mistral-7B / Mixtral-8x7B | WikiText-2 | 0.6 | SX | 128 | 6.56 | 3.23 | 3.2x |

Table 3: SpecExec inference speed on consumer GPUs with offloading, chat/instruct models, Llama 2 70B-GPTQ target model, $t = 0.6$, OpenAssistant dataset.

| GPU | Draft model | Budget | Gen. rate | Speed, tok/s | Speedup |
|---|---|---|---|---|---|
| RTX 4090 | | 256 | 13.46 | 5.66 | 8.3x |
| RTX 4060 | Llama 2-7B | 128 | 9.70 | 3.28 | 4.6x |
| RTX 3090 | | 256 | 14.3 | 3.68 | 10.6x |
| RTX 2080Ti | ShearedLlama-1.3B | 128 | 7.34 | 1.86 | 6.1x |

We also explore the relationship between the inference speed and the draft tree size. A larger draft budget allows for a greater number of tokens to be generated per step (see Figure 5 (left)). However, beyond a certain size threshold (hundreds or thousands of tokens, depending on the model and GPU), the time required for generation increases at an accelerating rate (see Figure 1 (right)). Consequently, the optimal draft tree size is typically smaller than the size that maximizes the token acceptance rate. According to our findings, displayed in Figure 5 (right), the optimal draft tree size is 128–512 for SpecInfer and 1024–2048 for SpecExec for the A100 GPU.

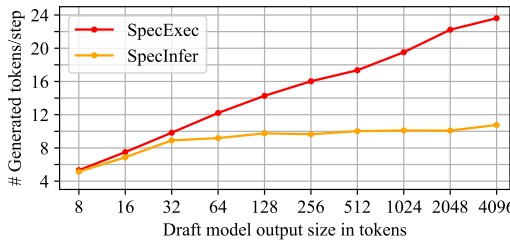 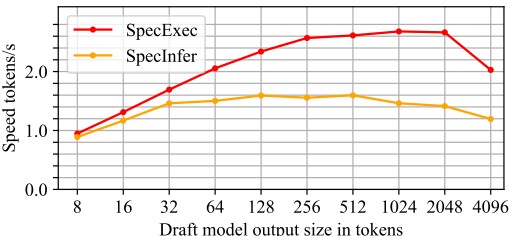

Figure 5: Acceptance counts (left) and generation speed (right) depending on the draft size. Llama 2-7B is used as the draft model, offloaded Llama 2-70B is the target model, MTBench dataset, t=0.6 and top-p=0.9. Results are obtained with an A100 GPU.

While this was not the primary focus area of our study, the SpecExec method can also deliver competitive speedups in inference without offloading; see Appendix G for sample results. Additional tests of SpecExec in generation with penalties show that the method is robust with such conditions: Appendix H provides the results of such an evaluation.

## 6 Conclusion and Future Work

In this work, we propose a method for fast inference of large models on consumer GPUs that unites the efficiency of offloading and speculative decoding in the large-budget setup. The resulting method, SpecExec, shows competitive performance in real-world experimental setups, demonstrating the possibility of running large models locally at the speed of interactive inference.

Although we developed an offloading system to utilize SpecExec in practical settings, the goal of our study was not to create the fastest possible implementation of local LLM inference. Achieving that goal relies on combining our approach with orthogonal performance improvements proposed in prior work, which is beyond the scope of this paper. Importantly, given the recent trends in hardware accelerators for deep learning, inference of large models may become increasingly more constrained by the memory bandwidth even for the fastest devices. Therefore, optimizing generation time with bandwidth constraints in mind is likely to grow more important in the future, and our work demonstrates a novel approach to that problem.

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

# A   Equivalence of Optimal Tree Search to Shortest Path Search

We can formulate this problem as follows:

$$\arg\max_{\tau \in \mathcal{T}^K} \sum_{x_i \in \tau} P_{\text{reach}}(x_i|\tau) \cdot P_{\text{accept}}(x_i|\tau). \tag{1}$$

Here, $x_i \in \tau$ refers to a token within the draft tree, $\mathcal{T}^K$ is a set of all trees of $K$ tokens and $P_{\text{reach}}(x_i|\tau)$ is the probability that the Speculative Execution verification phase accepts the full prefix $x_0, \ldots, x_{i-1}$ along the draft tree and considers sampling $x_i$ next. Finally, $P_{\text{accept}}(x_i|\tau)$ is the probability that the token $x_i$ will be accepted *if* it is reached during verification. Both $P_{\text{reach}}$ and $p_{\text{accept}}$ depend on the target model probabilities $P(x_{t+1}|x_{0:t}, \theta_{\text{target}})$, which cannot be accessed in the drafting phase. Instead, we use the draft model to approximate the target model probabilities as follows:

$$P_{\text{reach}}(x_i|\tau) \approx \prod_{x_t \in \pi(x_i, \tau)} P(x_t|\pi(x_t, \tau), \theta_{\text{draft}})$$
$$P_{\text{accept}}(x_i|\tau) \approx P(x_i|\pi(x_i, \tau), \theta_{\text{draft}}), \tag{2}$$

where $\pi(x_i, \tau)$ is the path in $\tau$ from root to $x_i$, excluding $x_i$ itself. From the LLM perspective, $\pi(x_i, \tau)$ is the prefix for a token $x_i$ within the draft tree. If we multiply the two expressions as per Equation 1, we get the cumulative probability of a sequence $\pi(x_i, \tau) \oplus x_i$, where $\oplus$ is concatenation.

$$\arg\max_{\tau \in \mathcal{T}^K} \sum_{x_i \in \tau} \prod_{x_t \in \pi(x_i, \tau) \oplus x_i} P(x_t|\pi(x_t, \tau), \theta_{\text{draft}}) \tag{3}$$

Since token probabilities cannot be greater than 1, the cumulative probability of $\pi(x_i, \tau) \oplus x_i$ cannot exceed the cumulative probability of all tokens in $\pi(x_i, \tau)$. Therefore, if a token $x_i$ is among the $K$ most likely tokens, every token in $\pi(x_i, \tau)$ is also a part of the solution. Using this property, we can simplify Equation 3 as finding top-$K$ most likely prefixes, since they are guaranteed to form a tree. Formally speaking, the optimal tree consists of K tokens with the highest cumulative probability:

$$\arg\,\text{top}\,K_{x_i} \prod_{x_t \in \pi(x_i, \tau) \oplus x_i} P(x_t|\pi(x_t, \tau), \theta_{\text{draft}}) \tag{4}$$

This is similar (but not equivalent) to the standard beam search algorithm for neural sequence models Graves [2012], Boulanger-Lewandowski et al. [2013], Sutskever et al. [2014]. The main difference is that beam search looks for complete sequences, while we need a tree of partial drafts. However, using beam search instead of solving Equation 3 directly leads to suboptimal drafts (see Appendix F for details).

Instead, we solve Equation 4 by reformulating it as a special case of the shortest path search problem. More specifically,

$$\arg\max_{x_i} \prod_{x_t \in \pi(x_i, \tau) \oplus x_i} P(x_t|\pi(x_t, \tau), \theta_{\text{draft}}) =$$
$$= \arg\max_{x_i} \sum_{x_t \in \pi(x_i, \tau) \oplus x_i} \log P(x_t|\pi(x_t, \tau), \theta_{\text{draft}}) = \tag{5}$$
$$= \arg\min_{x_i} \sum_{x_t \in \pi(x_i, \tau) \oplus x_i} -\log P(x_t|\pi(x_t, \tau), \theta_{\text{draft}}).$$

Note that every term in that sum is non-negative, (since $-\log P(x_t|\pi(x_t, \tau), \theta_{\text{draft}}) \geq 0$), which makes this equivalent to a single-source shortest path (SSSP) problem for finding paths to $K$ nearest nodes in a graph with non-negative edge weights. Normally, this problem can be solved by running the Dijkstra algorithm for $K$ steps. However, in practice, running the algorithm for $K$ sequential steps is inefficient on modern highly parallel hardware, especially in our setting with very large drafts. To alleviate this problem, we use a modified parallel Dijkstra algorithm, which expands $B > 1$ nodes on every iteration and keeps track of $K$ nearest nodes in a priority queue. We describe this formally in Algorithm 2.

In the worst case, this algorithm still makes up to $K$ steps if the solution to Equation 3 is a single "stem" B tokens long. However, the actual number of steps is significantly lower, often slightly above the lower bound $\lceil B/K \rceil$. In the practical GPU implementation, we also limit the maximum **d**epth with a parameter $D$. The purpose of D is to limit the edge case where the draft model is very confident about the next token, and thus the solution to Equation 3 is a single sequence of length K. For this edge case, Algorithm 2 will take long to generate a sequential draft, most of which will later be discarded if the draft model makes even one mistake.

## B   SpecExec Algorithm Diagram

Figure 6 displays a block diagram that outlines the key steps of the SpecExec algorithm.

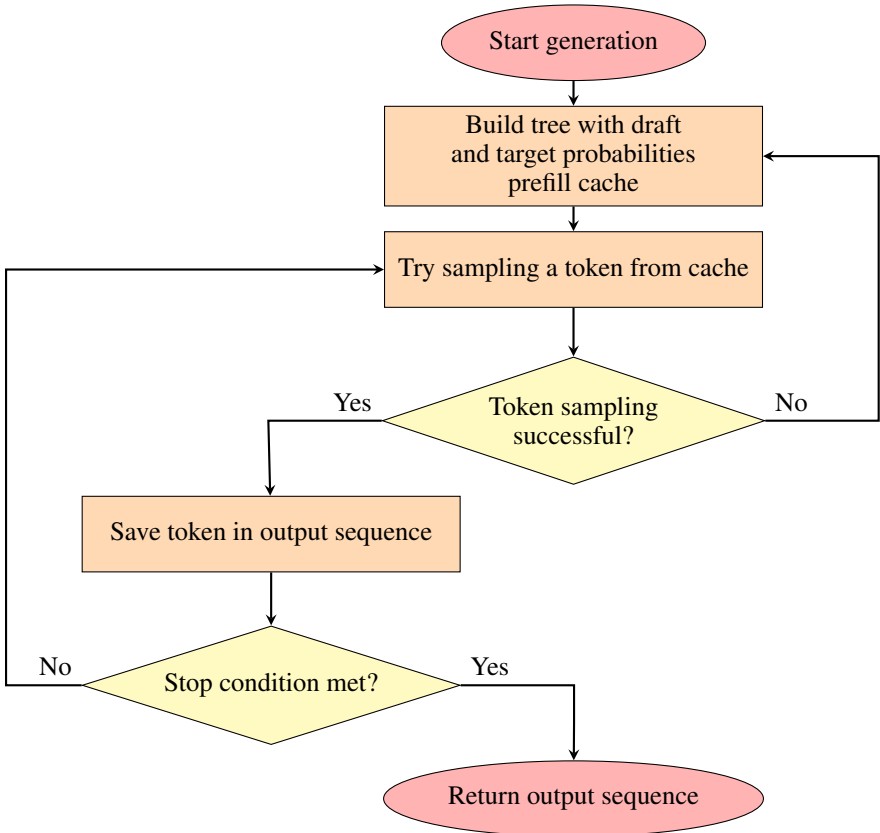

Figure 6: A high-level overview of the SpecExec algorithm.

## C   Parallel Graph Search

There are dozens of works that study practical implementations of parallel shortest path search and SSSP in particular. One line of work proposes inexact search algorithms that use an approximate priority queue to improve the performance of SSSP: Nguyen et al. [2013] proposes a queue with an integer metric, and Zhang et al. [2020] adds bucket fusion to reduce the synchronization overhead..

A significant effort was dedicated to efficient shortest-path search on GPUs, among others. Harish and Narayanan [2007] proposes a GPU-efficient SSSP that outperforms sequential CPU computations. Davidson et al. [2014], Wang et al. [2016] compare several SSSP variants for GPUs. Iacono et al. [2019] adapts priority queues to run efficiently on GPU and uses the resulting data structure to accelerate SSSP.

Many works on parallel graph search focus on the distributed setting Malewicz et al. [2010], Zhu et al. [2016], Besta et al. [2017], addressing communication and synchronization overheads. Finally,

a large body of work studies the theoretical properties of parallel SSSP, including Ullman and Yannakakis [1990], Klein and Subramanian [1997], Cohen [1993], Shi and Spencer [1999], Cohen [2000], Spencer [1997], Meyer [2001], Blelloch et al. [2016].

## D   Additional Implementation Details

Our system design follows the following loop:

1. Load the draft model onto GPU memory and generate a draft tree;
2. Load the main model (several layers at a time, if using offloading) to compute probabilities for the draft tree tokens;
3. Choose the accepted tokens following the verification procedure.

When running the main model, we process all draft tokens in parallel by constructing a merged attention mask, similar to Miao et al. [2023]. We prefetch the first few layers of the main model during speculation to speed up the procedure. We also load subsequent LLM layers in parallel, while the previous layers compute their activations, as described in Pudipeddi et al. [2020], Aminabadi et al. [2022].

Finally, we keep the past key/value caches of both draft and main models in GPU memory at all times. We chose this because most modern language models use grouped-query attention Ainslie et al. [2023], making caches relatively small for short prompts. When dealing with longer prompts or smaller GPU memory, one can reduce memory usage by offloading these KV caches into RAM. The draft model caches are only needed on GPU during the first stage when generating the draft tokens. In turn, the main model caches can be loaded alongside their transformer layers.

The optimal implementation of this algorithm is slightly different depending on the hardware configuration. Running SpecExec on a system with GPU with RAM offloading works best with relatively fewer draft tokens, while longer offloading (to SSD or when using float16 precision weights) works best with larger token budgets.

As for the quantization scheme, while there are better quantization algorithms Lin et al. [2023], Dettmers et al. [2023], Chee et al. [2023], we chose GPTQ since it is popular among practitioners. Still, we believe that our experimental results will generalize to other quantization algorithms. In addition to the main model, we also quantize the draft (7B) model using the same GPTQ algorithm. The optimal choices of the quantization methods will vary as new methods or faster implementations appear.

The experiments were mainly performed using A100 GPUs (unless specified otherwise), but may be easily reproduced using other GPUs. Note that while A100 has 80GB VRAM, we did not keep any layers in VRAM in order to keep the VRAM use to minimum and emulate performance of GPUs like RTX4090 or L40. A s a result, the observed VRAM use requirements with offloading was in 12–22 GB range for experiments with draft trees up to 2048 tokens when using Llama-2-70B RAM offloading. Naturally, keeping some of the layers constantly in VRAM would increase both baseline and the model performance.

The offloading experiments require sufficient RAM to hold whole model. In case of Llama-2-70B in 16 bit, this is at least 140 GB, but in practice 192 GB would be recommended to fit the draft model, caches and memory of other processes. Our code is based on industry standard PyTorch Paszke et al. [2019] and Transformers Wolf et al. [2019] libraries.

## E   Ablation: Acceptance with Different Draft Models

In Section 5.2, we evaluate SpecExec and SpecInfer with 7B draft models based on the observations about their coverage probabilities. Here, we further compare these models in terms of the number of accepted tokens for different SpecExec batch sizes. We report the results of this comparison in Figure 7 using the same OpenAssistant dataset as in the main experiments using the recommended temperature (0.6) and nucleus size (0.9). Similarly to Figure 2, the 7B model significantly outperforms both `JackFram/llama-160m` and TinyLlama 1.1B Chat. This is true both for the original 7B model and the one quantized to 4 bits with GPTQ. Curiously, the full unquantized 13B model still obtains

slightly more accepted tokens, though at the cost of 26GB memory footprint that is inaccessible to modern consumer GPUs.

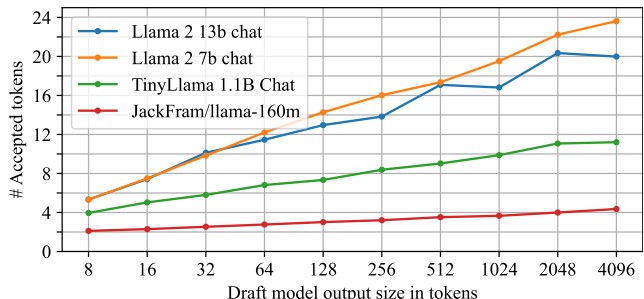

Figure 7: Number of accepted tokens as a function of the draft size $B$ for the Llama 2-70B Chat target model and different draft models.

## F    On The Suboptimality of Beam Search

In our preliminary experiments, we tried to construct the optimal draft tree using top-k beam search decoding Graves [2012]. However, we observed that the algorithm performed significantly worse than expected and often plateaued as we increased the maximum beam search length. Here, we describe the analysis of this problem that eventually led us to Algorithm 2.

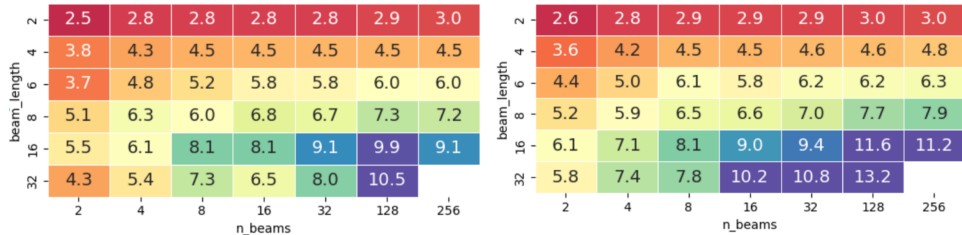

Figure 8: The average number of accepted tokens per speculation and verification phases as a function of beam size and maximum length. The measurements are obtained on OpenAssistant conversations (Left) and WikiText-2 articles (Right) for running with recommended generation parameters (temperature 0.6, top-p 0.9). **(Left)** standard beam search decoding, **(Right)** beam search without pruning out-of-beam tokens.

Figure 8 (left) reports a grid where each cell is the number of accepted tokens for a version of SpecExec that uses beam search instead of parallel SSSP. The horizontal grid axis corresponds to beam size (also known as the number of beams), and the vertical axis depicts maximum length within a beam. The left grid shows standard beam search decoding that returns beam size most likely sequences. In turn, the right grid uses a modified search algorithm that starts the same way as beam search but does not prune any partial hypotheses that did not make it into the final beam.

As we can see, standard beam search decoding is suboptimal for SpecExec in the sense that it can be outperformed with trivial modifications. In turn, Algorithm 2 is a generalization of the version on the right that does not need to be manually tuned for length and width, but instead expands the graph optimally to maximize the coverage probability.

## G    Application to in-memory inference

While this was not the primary focus area of our research, the SpecExec method can also deliver measurable speedups in inference without offloading. While these speedups are less impressive than those for offload settings, they are still competitive when compared to recent works such as Chen et al. [2024a].

Table 4: SpecExec Inference speed without offloading, A100 GPU.

| Draft / Target models | Dataset | t | Method | Budget | Gen. rate | Speed, tok/s | Speedup |
|---|---|---|---|---|---|---|---|
| SL-1.3B / Vicuna-33B | OASST-1 | 0.6 | SX | 128 | 5.33 | 31.6 | 2.15x |
| | OASST-1 | 0 | SX | 128 | 5.4 | 32.94 | 2.24x |
| | C4 | 0.6 | SX | 128 | 5.1 | 33.3 | 2.26x |
| | C4 | 0 | SX | 128 | 5.36 | 35.62 | 2.42x |
| | WikiText-2 | 0.6 | SX | 128 | 4.87 | 30.19 | 1.90x |
| | WikiText-2 | 0 | SX | 128 | 5.24 | 33.15 | 2.08x |

## H Drafting penalty effects

To verify the method's robustness, we ran a series of experiments with penalties excluding the use of specific tokens. The same penalty scheme was applied to both draft and target models, and the expectation is that the models should be able to run SpecExec effectively. To verify this claim, we ran a series of experiments with penalties excluding use of fewer or more tokens. For these experiments, we penalized all tokens that start from the letter "r" (left) or all tokens that contain the letter "r" (right). Here we used the Llama 2-7B Chat target model with TinyLlama-1.1B Chat draft model, t=0.6, p=0.9, MT-Bench dataset.

The results of these experiments can be found in Figure 9. We found that our method's performance (measured in terms of accepted tokens per iteration) stays stable only with lightweight penalties, yet heavier penalties reduce the absolute speedups. Looking at the generated samples, we observed that while with lighter penalties, the model is able to work around the restrictions and generate reasonable text, with heavier penalties the quality deteriorated as the model skipped or replaced tokens seemingly at random. Stronger penalties affect the quality of the generated text and naturally make the task harder for the draft model. Thus, we attribute the lower performance with a heavy penalty to this perplexity increase rather than to the penalty directly.

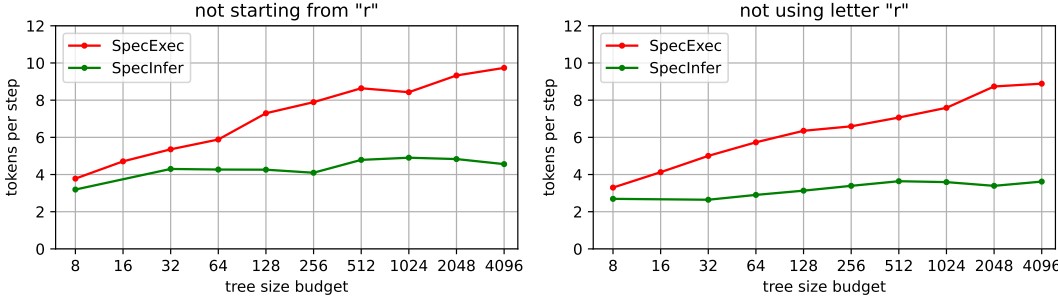

Figure 9: Acceptance rate in generation with token penalty: "don't start words with "R"" (left) and "don't use the letter "R"" (right); Llama 2 7B Chat (+ TinyLlama-1.1B Chat draft) model (t=0.6, p=0.9), MT-Bench dataset. The rest of the experimental configuration is the same as in Figure 1.

