# OpenReview forum: "SpecExec: Massively Parallel Speculative Decoding For Interactive LLM Inference on Consumer Devices"
_NeurIPS.cc/2024/Conference — NeurIPS 2024 poster_

### Official Review · Reviewer_PkkW · 2024-06-20

**Soundness:** 3
**Presentation:** 3
**Contribution:** 3
**Rating:** 5
**Confidence:** 3

**Summary:**

This paper presents a novel speculation decoding method called SpecExec, designed to improve the performance of large language model (LLM) inference on consumer-level GPUs with offloading scenario. The main contribution is the application of speculative execution, a technique from CPU architecture, to the speculation decoding of LLMs. SpecExec leverages a draft model to construct a large draft tree of potential continuations, which are then verified in parallel by the target model. This approach aims to mitigate the inefficiencies of existing speculative decoding methods, particularly when dealing with a large number of draft tokens. The experiments result demonstrate that SpecExec achieves up to 18.7x speedup.

**Strengths:**

- This paper tackles specific and clear setup. Since offloading  is the main bottleneck of LLM inference on consumer-level GPUs, using speculative decoding with many draft tokens is much more preferable.
- SpecExec is robust for non-deterministic request (i.e., t = 0.6, top_p = 0.9)
- As the SSSP problem is one of the well-studied problems in the CS domain, it may be easy to adopt other variants of SSSP to generate different draft trees (extendability).

**Weaknesses:**

- Although the paper targets consumer-level GPUs, the experiment results show that SpecExec achieves the highest speedup with the A100 GPU, one of the most popular datacenter-level GPUs.
- Token throughput is still slow because of the offloading. For A100 GPU, using 4-bit quantized 70B model (AWQ, GPTQ, ...) and without speculative decoding will be more practical.
- SpecExec seems plausible only for single request with single user.

**Questions:**

- How much speedup does SpecExec gain without RAM offloading? For example, Llama2-70b chat GPTQ model on an A100 GPU. I am just curious
- So, if I understood correctly, sampling parameters such as t = 0.6, p = 0.9 do not affect the draft tree, right? (deterministic tree)
- In Figure 3 and 4, random sampling results are comparable with greedy sampling on the MTBench dataset while the C4 dataset shows lower hit ratio on random sampling. Does this difference come solely from the difference between datasets?
- Can SpecExec achieve high performance gain on token penalty scenarios? For example, repetition penalty, presence penalty, and frequency penalty.

**Limitations:**

No additional limitations regarding societal impact.
For technical limitations, I stated in the weakness section.

---

> ### Author Rebuttal · Authors · 2024-08-07
>
> We thank the reviewer for the feedback and questions. We are glad that the reviewer appreciates the practical impact of our work for LLM inference on consumer GPUs. Below we address the questions to the best of our ability:
>
> > Although the paper targets consumer-level GPUs, the experiment results show that SpecExec achieves the highest speedup with the A100 GPU, one of the most popular datacenter-level GPUs.
>
> We do indeed show high speed-ups on A100 GPUs, but we respectfully insist that this is not a weakness. Our main setup is still consumer GPUs — see Table 3 where we show significant speed-ups on consumer GPUs. Attaining speedup on a datacenter-grade A100 GPU is less challenging, but has a positive side-effect. We are glad that our approach can also accelerate datacenter applications. In light of recent LLM developments, this could allow datacenter users to run extremely large models (Nemotron-340B, Llama 3.1 405B) that do not fit on a single datacenter-grade GPU.
>
> > Token throughput is still slow because of the offloading. For the A100 GPU, using a 4-bit quantized 70B model (AWQ, GPTQ, ...) and without speculative decoding will be more practical.
>
> In short, our method allows to use even less VRAM and fully preserve the original model quality.
>
> Quantization methods like GPTQ are indeed trying to solve a similar problem: to run LLMs with lower VRAM utilization. Quantized models can deliver decent compression at a slight cost in accuracy. Still, our method opens additional applications: (a) even lower vRAM requirements. Llama 70B GPTQ in 4 bit requires ~35 GB of VRAM, while our method allows it to fit into under 20GB (with Llama 7B draft model) or possibly even under 12GB if using a less capable draft model. (b) some users may insist on non-quantized models for applications requiring every last bit of performance or reproducing results of the original models.
>
> Furthermore, the quantized models may be offloaded and accelerated using our method, offering further speedups. We use GPTQ target models for some experiments reported in Tables 1 and 2, achieving an acceleration of up to 9x (against autoregressive inference) due to faster offloading.
> While there indeed are setups that SpecExec is not optimal for, our paper focuses on a narrow but practically important task of inferencing models with offloading, where we achieve significant speed-ups.
>
> > SpecExec seems plausible only for single request with single user
>
> You are correct, and this is our target application setting. While our algorithm may be adopted for batch applications, it would require significantly more VRAM which is not what our typical user may have. Our objective is to give users of less capable GPUs the ability to run modern LLMs with decent quality and usable speed.
>
> > How much speedup does SpecExec gain without RAM offloading?
>
> The in-memory (no offloading) setting allows relatively less time for speculation compared to offloading-scale timings. Nevertheless, our method demonstrates decent speedups of up to 2.4x with Vicuna-33b model: please refer to Appendix F for experiment results in this setup.
>
> > sampling parameters such as t = 0.6, p = 0.9 do not affect the draft tree, right? (deterministic tree)
>
> In short, the SpecExec draft tree is not deterministic, but its shape in our implementation is not dependent on generation temperature.
> We experimented with applying the same sampling temperature to the draft model when constructing the tree, however, we found that keeping the draft model always sampling at temperature 1.0 gives the most consistent results. Our algorithm builds the tree dynamically, selecting the best new nodes in order to maximize the expected length of the text continuations covered by the tree.
>
> > Figure 3 and 4, random sampling results vs greedy for MTBench and C4/ - Does this difference come solely from the difference between datasets?
>
> The non-chat C4 dataset has a higher inherent text entropy, so when generation temperature grows, the model gets a wider choice of reasonably probable tokens, and matches between top choices of the draft and target models become relatively less frequent. Still, our algorithm outperforms SpecInfer in this scenario by a large margin.
>
> >Can SpecExec achieve high performance gain on token penalty scenarios?
>
> In theory, the algorithm could work with any penalty that modifies the model probabilities. If the same penalty scheme is applied to both draft and target models, the models should be able to run SpecExec effectively.
> To verify this claim, we ran a series of experiments with penalties excluding use of fewer or more tokens.  The results of these experiments can be found in **Figure 2 in the PDF attachment**. More specifically, we penalize all tokens that start from the letter “r” (Figure 2 left) or all tokens that contain the letter “r” (Figure 2 right). We use the same setup as in the previous experiments (Figure 1).
>
> We found that our method’s performance (measured in terms of accepted tokens per iteration) stays stable only with lightweight penalties, while heavier penalties reduce the absolute speedups. Looking at the generated text, we observed that while with lighter penalties the model is able to work around the restrictions and generate reasonable text, with heavier penalties the quality deteriorated as the model skipped or replaced tokens seemingly at random. Stronger penalties affect the quality of the generated text and naturally make the task harder for the draft model. Thus, we attribute the lower performance with a heavy penalty to this perplexity increase rather than to the penalty directly.
>
> We hope that the additional experiments and explanations help address your concerns and answer questions. If the reviewer has any further suggestions on improving the paper, we encourage them to include them in the OpenReview, e.g. by editing the review.

---

### Official Review · Reviewer_BH6v · 2024-07-05

**Soundness:** 3
**Presentation:** 1
**Contribution:** 3
**Rating:** 5
**Confidence:** 3

**Summary:**

In this paper, the authors propose a speculative decoding technique, SPECEXEC (Speculative Execution), that can generate up to 20 tokens per iteration for the LLaMA-2 70B model. SPECEXEC enables the LLaMA-2 70B model to run on a consumer GPU by a parallel decoding strategy. The offloading scheme can process thousands of tokens using the same latency as that of one token. Therefore, the offloading scheme can be naturally used in supporting speculative decoding. However, prior speculative decoding techniques do not scale with a large batch of draft tokens. SPECEXEC uses a smaller draft model to built a tree of best tokens by single-source shortest path (SSSP) searches, and then processes best tokens with the target model by offloading. SPECEXEC also includes a new algorithm to determine which tokens are accepted.

**Strengths:**

1. The paper works on an important topic.
2. The paper includes enough supporting data.

**Weaknesses:**

1. The paper is not well-written. SPECEXEC is built on a important observation that the offloading of the target model can process thousands of tokens using the same latency as that of one token. SPECEXEC suffers from a lower acceptance rate for individual tokens. For a small number of draft tokens, SpecExec is less efficient than speculative decoding. However, for a large enough number of draft tokens, SpecExec obtains a better tree and increases the accept rate. SPECEXEC adopts the offloading technique to mitigate the processing of the  large enough number of draft tokens. Unfortunately, the authors did not justify the applications or user cases that must use the offloading schemes.

**Questions:**

1. What type of applications are more suitable for SpecExec? The applications have not enough server-level GPUs?

**Limitations:**

no limitation.

---

> ### Author Rebuttal · Authors · 2024-08-07
>
> We thank the reviewer for their feedback; we are glad that they appreciate our experimental results. Below, we do our best to address the concerns and answer questions.
>
> >The paper is not well-written.
>
> We are eager to improve our writing and respectfully ask the reviewer to suggest specific improvement areas (e.g., in the updated review). While there are improvement areas, other reviewers found our presentation to be good (3), and we hope that the remaining issues can be fixed. We will pay significant attention to the writing and presentation style when developing the final version of the paper.
>
> > SPECEXEC suffers from a lower acceptance rate for individual tokens.
>
> This is technically correct, but it is not a weakness of SpecExec. As we discuss in Lines 148-159, we deliberately use a simpler acceptance algorithm to lift the constraints on the draft tree structure. In turn, this allows us to dynamically build the best tree for a given budget, unlike SpecInfer which requires a specific sampling distribution.
>
> The algorithm may occasionally exhibit a lower acceptance rate **for individual tokens**, when compared to SpecInfer. However, in practice, the average individual acceptance rate is not much different. This choice allows for superior global acceptance rate and speedup which are the key metrics important to the algorithm users.
>
> > For a small number of draft tokens, SpecExec is less efficient than speculative decoding. However, for a large enough number of draft tokens, SpecExec obtains a better tree and increases the accept rate.
>
> SpecExec is less effective than traditional speculative decoding only for cases with very small trees (1-4 tokens). In our main experiments, SpecExec is very close to SpecInfer for small budgets of up to 32 tokens draft tokens, but still slightly outperforms it. Please refer to Figures 3 and 4, where SpecExec attains better results for all considered token budgets. We attribute this to the fact that SpecExec picks the best tokens (most probable tree cumulatively), while SpecInfer and similar methods have to sample draft tokens randomly in order to maintain output distribution. However, the degree to which SpecExec outperforms SpecInfer is smaller because there is less potential to construct better draft trees.
>
> > the authors did not justify the applications or user cases that must use the offloading schemes - add details and examples. What type of applications are more suitable for SpecExec? The applications have not enough server-level GPUs?
>
> In general, our approach can be helpful for LLM inference in any resource-constrained environment. One such environment is running modern LLMs with tens of billions of parameters on desktop hardware with consumer-grade GPUs.  Another important application is model inference on smartphones where even relatively small models need to be offloaded. Moreover, our approach can be applied to datacenter-grade GPUs running extremely large models (Nemotron-340B, Llama 3.1 405B).  Finally, there is a potential application to run extremely large models in a distributed environment, where the communication delays add latency to the base model, which is similar to the delays that stem from offloading.
>
> We hope that our reply alleviates your concerns and welcome further suggestions in the updated review.

---

### Official Review · Reviewer_ph7U · 2024-07-09

**Soundness:** 3
**Presentation:** 3
**Contribution:** 3
**Rating:** 6
**Confidence:** 4

**Summary:**

The authors present a method to improve the efficiency of speculative decoding on consumer-grade hardware. The technique addresses the inefficiencies of existing speculative decoding approaches when applied to devices with limited GPU memory, necessitating parameter offloading to RAM or SSD. SpecExec leverages a powerful draft model to create a speculative tree of potential token continuations, which the target model then verifies. This approach enables the processing of multiple tokens in parallel, significantly improving the inference speed on consumer GPUs.

**Strengths:**

The paper provides a solid empirical analysis of key inefficiencies of speculative decoding and presents a well-founded solution. The paper is written clearly with solid motivation and method sections.

The empirical results that the authors provide are impressive, showing consistent speedup over SpecInfer on the number of generated tokens per step.

In addition to results on generated tokens per step, the authors also present an impressive speedup on tokens per second, showing that the improvement achieved at generated tokens per step can be translated to real wall-clock speedup.

**Weaknesses:**

1. Some experiment results are a bit confusing. For instance, the captions in Figure 3/4 say Generation rate vs draft size for Llama 2-7B/70B models, but it's not clear where the 7B model numbers are. The main text states the plots are for 70B models.

2. The authors claim they have results on both base and chat models. But it's not clear from the tables and figures, which are chat models and which are base models. And further, whether this would lead to different results/conclusions.

**Questions:**

My questions are mainly regarding the experiment details discussed above in the weakness section.

**Limitations:**

Yes, the authors adequately addressed the limitations and, if applicable, potential negative societal impact of their work.

---

> ### Author Rebuttal · Authors · 2024-08-07
>
> We thank the reviewer for the well rounded review of the paper and address the questions below:
>
> > Some experiment results are a bit confusing. For instance, the captions in Figure 3/4 say Generation rate vs draft size for Llama 2-7B/70B models, but it's not clear where the 7B model numbers are. The main text states the plots are for 70B models.
>
> In these figures, we use the 7B model **as a draft model**, whereas the 70B is the target model.
> Since the algorithm employs a separate draft model to speculatively develop a continuation tree, each experiment uses two models. Namely, we use a 7B draft model to develop a tree and a 70B target model for verification. We agree that the captions to the figures could be more detailed and will update them.
> To further alleviate the reviewer’s concern, we have conducted an **additional set of experiments where the 7B model is the target model**, as suggested. For this experiment, we pair the 7B target model with TinyLlama 1.1B as the draft model. The rest of the experiment configuration is the same as in Section 5.2. The results can be found in **Figure 1 in the PDF attachment**.
>
> > The authors claim they have results on both base and chat models. But it's not clear from the tables and figures, which are chat models and which are base models. And further, whether this would lead to different results/conclusions.
>
> Following the examples of other speculative decoding papers, we report results separately for the two model classes. For most figures and tables, we use the word “chat” in the caption to indicate the results of the instruction-trained models, with a few omissions that we will rectify. Specifically, Figures 2, 4, 5, 6 and Tables 1, 3 and 4 are for chat models. Figures 1, 3, 7 and Table 2 provide results for non-chat models. We missed this caption in Figure 2, where we use chat models.
>
> Overall, we noticed that the algorithm demonstrates similarly high performance with both chat and base models. We thank the reviewer for bringing the missing captions to our attention. In the final version of the paper, we will add clearer labels to make the figures easier to follow.

---

### Official Review · Reviewer_fKWn · 2024-07-11

**Soundness:** 3
**Presentation:** 3
**Contribution:** 4
**Rating:** 7
**Confidence:** 3

**Summary:**

This work introduces SpecExec, an improved speculative decoding method.

By constructing a better draft token tree and refining the process of verifying tokens, SpecExec significantly increases the number of tokens accepted in a single verification while producing exactly the same outputs as sequential sampling from the target model.

Additionally, SpecExec leverages the capability of modern offloading engines to dispatch model parameters in parallel with computation, successfully achieving effective acceleration under offloading configurations.

Experiments demonstrate that SpecExec can run a 70B model on consumer-grade GPUs and achieve up to 10.6 times acceleration.

**Strengths:**

1. SpecExec uses a modified version of Dijkstra’s algorithm to create better token trees, significantly increasing the number of tokens accepted in a single verification.

2. SpecExec views the process of parallel verification of candidate tokens as looking them up in a speculative “cache”, ensuring that the output is identical to that of sequential sampling, while speculative decoding only guarantees correct overall sampling probabilities.

3. SpecExec has demonstrated excellent acceleration in experiments, significantly outperforming SpecInfer and showcasing superior performance.

**Weaknesses:**

1. While SpecExec achieves significant performance acceleration in systems with offloading, it is unclear how this method performs in terms of acceleration in speculative inference systems that do not have offloading.

**Questions:**

1. In Line 14 of Algorithm 1, "next_probs := FORWARD(τ, θ_target)", how is forward inference performed on the draft tree? Is it the same method used in SpecInfer?

2. The paper lacks clear illustrations of the working process of SpecExec. Could additional diagrams be added to provide a more vivid explanation?

3. The paper mentions several articles related to speculative decoding but lacks an introduction to these works and does not explain the differences between them. Adding this background could help readers better understand this paper and it's contribution.

**Limitations:**

The authors have not adequately discussed the limitations of this paper.

---

> ### Author Rebuttal · Authors · 2024-08-07
>
> We thank the reviewer for the feedback and appreciate that they share our views on the algorithm performance factors. Below, we do our best to address the reviewer’s concerns and answer questions.
>
> >  It is unclear how this method performs in terms of acceleration in speculative inference systems that do not have offloading.
>
> We report experiments for the non-offloading setting in Appendix F (referenced in L287). The in-memory (no offloading) setting allows relatively less time for speculation compared to offloading. Nevertheless, our method demonstrates considerable speedups of up to 2.4x even in this setting.
>
> >  Line 14 of Algorithm 1, … how is forward inference performed on the draft tree?
>
> First, SpecExec uses the draft model to construct the draft tree in such a way that it covers the most probable text continuations. Next, in line 14 the algorithm runs the target model to get “true” probabilities in each of the tree nodes, which is done in a single forward pass, using a custom 4D “tree-attention” mask (L537-538).  This prefills a speculative “cache” of continuations to choose from  (See Alg.1 L6). Finally, we use these cached target model’s probabilities to generate final tokens.
>
> > Could additional diagrams be added to provide a more vivid explanation? (...)  Adding this background could help readers better understand this paper and its contribution.
>
> Thank you for the suggestion. We offer a limited overview of speculative decoding works in Section 2.1, but we agree that the paper would benefit from a more detailed introduction to the topic.
> We had ideas on more detailed background and extra illustrations, but chose instead to allocate more of our page limit to the algorithm and experiments. Should our paper be accepted, we intend to use the extra space to improve the intuitive diagram of the algorithm and extend the background section.

---

### Author Rebuttal · Authors · 2024-08-07

We thank the reviewers for taking the time to study our paper and providing valuable feedback. We are glad to notice that all four reviewers appreciate the practical speed-ups achieved by SpecExec and its positive impact on LLM accessibility. On the reviewers' suggestions, we implemented a few additional experiments described in our individual responses and in the attached PDF. The reviewers gave us a number of ideas on making the paper more polished and complete, which we intend to implement for the final version.  The individual reviewers' questions and concerns are addressed in the reviewer-specific rebuttal sections.

Additionally, we'd like to highlight these common themes in several reviews:

- our method's somewhat lower acceptance rate on the individual token level is not a problem but a necessary trade-off, allowing us to build arbitrarily shaped trees, specifically ones that include the most probable tokens in each generation.

- our algorithm is focused on applications with relatively high target model latency, which makes multiple speculation iterations by a reasonably capable draft model affordable. Such a setup allows building quite large trees (up to thousands of tokens) and delivering speedups in 15x-20x range. Using our method with already fast models in an in-memory setting will also provide significant speedups, but those would not stand out among the ones from competitive methods.


In the attached PDF, we included several additional experiments to help us answer some of the reviewers questions:
* **(ph7U)** Additional evaluations where the 7B LLM is used as the target model, with a smaller draft model;
* **(PkkW)** Experiments on speculative decoding with a token penalty, comparing SpecExec and SpecInfer.

---

### Decision · Program_Chairs · 2024-09-25

**Decision:**

Accept (poster)

**Comment:**

The paper has an Average Rating: 5.75 (Min: 5, Max: 7) and is slightly above the acceptance threshold. While there are various issues, all reviewers have given positive evaluations. Considering the following strengths, I recommend acceptance(poster).

1: SpecExec offers a method that enables large LLMs to operate efficiently on consumer-grade GPUs. Even on consumer hardware, it achieves more than a tenfold speed improvement when combined with offloading, making LLMs more accessible and practical by enhancing their accessibility and usability.

2: The proposed method consistently shows higher speed improvements compared to existing SpecInfer methods, especially when processing large numbers of tokens in parallel. Significant improvements in actual computation time have also been confirmed.

3: SpecExec extends existing inference methods by utilizing Dijkstra’s algorithm to construct more efficient token trees, allowing multiple tokens to be verified simultaneously. This significantly improves speed while maintaining the same output as sequential sampling.

4: The SpecExec algorithm leverages techniques adapted from CPU architecture, making it robust and applicable to a broader range of computing environments and models in the future. It is particularly well-regarded for its applicability to scenarios where inference speed is critical.